

# The potential of serum elabela levels as a marker of diabetic retinopathy: results from a pilot cross-sectional study

Muhammed Seyithanoğlu[1], Selma Meşen[2], Aysegul Comez[2],
Ali Meşen[2], Abdullah Beyoğlu[2], Yaşarcan Baykişi[1] and
Filiz Alkan Baylan[1]

[1] Department of Biochemistry, Kahramanmaraş Sütçü İmam University Faculty of Medicine, Kahramanmaraş, Turkey
[2] Department of Ophthalmology, Kahramanmaraş Sütçü İmam University Faculty of Medicine, Kahramanmaraş, Turkey

Corresponding author
Muhammed Seyithanoğlu,
dr.muh.seyit@gmail.com

## ABSTRACT

**Background:** The aim of this study is to examine the relationship between elabela (ELA), a recently identified peptide also known as Toddler and Apela, and diabetic retinopathy (DR). ELA, produced in various tissues, acts as a natural ligand for the apelin receptor (APJ). Upon reviewing the existing literature, only one study was found investigating ELA, one of the APJ ligands, in the pathogenesis of DR.

**Methods:** In our study the patient group comprising individuals diagnosed with type 2 diabetes mellitus (DM), categorized into three subgroups based on detailed fundus examination: those without DR (non-DR) ($n = 20$), non-proliferative DR (NPDR) ($n = 20$), and proliferative DR (PDR) ($n = 20$). A control group ($n = 20$) consisted of individuals without DM. Blood samples were collected during outpatient clinic admission to measure serum ELA levels, which were determined using a commercial ELISA kit.

**Results:** The age, sex, and body mass index of the between groups were similar ($p = 0.905$, 0.985 and 0.241, respectively). The HbA1c levels of the between DM subgroups were similar ($p = 0.199$). Serum ELA levels were 217.19 ± 97.54 pg/mL in the non-DR group, 221.76 ± 93.12 pg/mL in the NPDR group, 302.35 ± 146.17 pg/mL in the PDR group and 216.49 ± 58.85 pg/mL in the control group. While ELA levels were higher in DM patients compared to the control group, this elevation did not reach statistical significance. Further analysis dividing DM patients into subgroups (non-DR, NPDR, and PDR) revealed higher ELA levels in the PDR group compared to the other subgroups, but this increase was not statistically significant.

**Conclusion:** Despite the absence of a significant difference in our study, the identification of elevated ELA levels in the PDR group offers valuable insights for future investigations exploring the association between DR and ELA.

# INTRODUCTION

Diabetic retinopathy (DR) is a significant complication of diabetes, with its likelihood of occurrence increasing as the duration of the disease extends (*Pivari et al., 2019*; *Sasso et al.,*

*2022*). DR progresses through two phases: the non-proliferative phase (NPDR) and the proliferative phase (PDR). During NPDR, retinal artery changes such as microaneurysms and intraretinal hemorrhages are observed. In the PDR stage, new blood vessels form in the retina due to retinal ischemia (*van der Giet et al., 2015*). Hypoxia-induced signaling plays a central role particularly in advancing to the proliferative stage. Retinal ischemia induces the release of angiogenic factors like vascular endothelial growth factor (VEGF), which mediates pathological neovascularization and increases vascular permeability (*Wei et al., 2022*; *Duh, Sun & Stitt, 2017*).

The pathophysiology of DR is driven by complex mechanisms, including chronic hyperglycemia, vascular dysfunction, oxidative stress, and inflammation, which contribute to both neuronal and vascular damage in the retina (*Wei et al., 2022*). Understanding the interplay between these mechanisms is critical for identifying and validating biomarkers that may aid in detecting or monitoring DR progression. Various studies have investigated biomarkers linked to these processes to better understand DR progression. For example, VEGF has been extensively investigated for its involvement in hypoxia-induced neovascularization, whereas other biomarkers, including cytokines like interleukin-1β (IL-1β) and interleukin-6 (IL-6), have been associated with retinal inflammation (*Duh, Sun & Stitt, 2017*; *Uemura et al., 2021*; *Ferrara, Gerber & LeCouter, 2003*). The Epidemiology of Diabetes Interventions and Complications study revealed that only 11% of the risk for DR is attributable to HbA1c levels, indicating that 89% of the risk is associated with other diabetes-related factors (*Lachin et al., 2008*). This underscores the necessity of exploring novel circulating biomarkers, which could not only provide insights into the disease mechanisms but also inform innovative clinical strategies for DR management.

Elabela (ELA) is a newly identified peptide composed of 54 amino acids. Initially active during early development, ELA continues to circulate as a secreted hormonal peptide in adults. It is produced in various tissues and serves as a natural ligand for the apelin receptor (APJ) (*Pauli et al., 2014*). Before the discovery of ELA, apelin (APLN) was considered the sole ligand for the G protein-dependent APJ (*Zhang et al., 2018*). Recent research indicates that ELA and APLN share similar functions, both operating through the APJ to form a crucial signaling axis in various physiological processes (*Kuba et al., 2019*).

The apelinergic pathway has gained increasing attention as a potential therapeutic target in cardiovascular and metabolic diseases. Additionally, the APLN/APJ system is important in the pathogenesis of eye diseases and plays a role in the regulation of eye health by affecting the retina (*Vural et al., 2021*; *Bezci Aygun et al., 2020*; *Wu, Chen & Li, 2017*). While APLN has been extensively studied in this context, ELA's role in eye diseases remains underexplored. Preliminary evidence suggests that ELA exerts beneficial cardiovascular effects, such as promoting blood vessel relaxation and reducing blood pressure (*Marsault et al., 2019*). It has been shown that ELA treatment improves acute kidney injury, serum ELA levels decrease in diabetic kidney disease (DKD), a complication of diabetes, and are associated with the severity of the disease and albuminuria (*Chen et al., 2017*; *Onalan et al., 2020*; *Zhang et al., 2018*). Given ELA's functional similarity to apelin in cardiovascular and metabolic systems and its involvement in the apelinergic pathway, it is plausible that ELA may contribute to DR-related processes such as hypoxia-driven

neovascularization and retinal vascular dysfunction. In this context, the present study aims to measure serum ELA levels in patients with DR, providing insights into its potential as a biomarker for disease progression. Understanding ELA's role could pave the way for new opportunities in early detection and targeted therapeutic strategies for DR.

A review of the literature revealed only one study investigating the relationship between ELA levels and DR. This study, presented as a poster at the 82nd Scientific Sessions of the American Diabetes Association, categorized type 2 diabetes mellitus patients into three groups: non-DR, NPDR, and PDR. The study reported that ELA levels decreased progressively from the non-DR group to the PDR group, with statistically significant differences among the three groups (Gu et al., 2022).

The limited availability of data underscores the need for further exploration of ELA's role in DR. By investigating ELA as a potential biomarker for disease progression, the current study aims to address this gap and contribute to the growing body of knowledge in the field.

## MATERIALS AND METHODS

This study was conducted at a tertiary Training and Research Hospital and was approved by the Clinical Research Ethics Committee of Kahramanmaraş Sütçü Imam University Faculty of Medicine (Decision date: October 17, 2023; Session number: 2023/17; Decision number: 04). The research adhered to the principles of the Declaration of Helsinki, and written informed consent was obtained from all participants. The study was carried out between October 2023 and January 2024.

### Study design and data source

This study was designed as a cross-sectional investigation to provide a comprehensive snapshot of serum ELA levels in DR patients compared with non-diabetic controls. This study included volunteers aged 40–80 years who had undergone cataract surgery. The patient group consisted of individuals diagnosed with type 2 diabetes mellitus (T2DM) based on the 2024 American Diabetes Association (ADA) diagnostic criteria (American Diabetes Association Professional Practice Committee, 2024). These patients were further classified into three subgroups according to findings from detailed fundus examinations: non-DR (no diabetic retinopathy) ($n = 20$), NPDR ($n = 20$), and PDR ($n = 20$). The control group ($n = 20$) comprised individuals without T2DM. DR classification followed the International Clinical Disease Severity Scale for DR, developed using data from the Wisconsin Epidemiological Study of Diabetic Retinopathy (WESDR) and the Early Treatment of Diabetic Retinopathy Study (ETDRS) (Wu et al., 2013).

**Non-DR Group:** Patients with no significant diabetic fundus abnormalities.
**NPDR Subclassification:**
**Mild NPDR:** Few microaneurysms.
**Moderate NPDR:** Microaneurysms, intraretinal hemorrhages, or venous beading.
**Severe NPDR:** Defined by the 4:2:1 rule, which includes hemorrhages in four quadrants, venous beading in two or more quadrants, and intraretinal microvascular abnormalities in at least one quadrant.

**PDR Group:** Characterized by optic disc or retinal neovascularization, iris or iridocorneal angle involvement, vitreous hemorrhage, or tractional retinal detachment.

Participants were screened *via* biomicroscopic dilated fundus examinations and grouped accordingly. Two experienced ophthalmologists (S.M. and A.M.) conducted the evaluations independently, and disagreements were resolved by a third ophthalmologist (A.B.).

## Exclusion criteria

Participants were excluded if they had type 1 DM, glaucoma, uveitis, traumatic cataracts, retinal vascular occlusions, retinal detachment, degenerative myopia, prior eye diseases, intraocular surgery, or eye trauma. Other exclusions included a history of systemic autoimmune or inflammatory diseases, recent intravitreal dexamethasone implantations (within 6 months), anti-VEGF treatments (within 3 months), or laser photocoagulation (within 3 months).

Medical histories were verified through electronic health records and direct patient interviews to ensure accurate documentation of exclusion criteria; additionally, physical examinations and laboratory records were reviewed to reduce potential bias.

## Data collection

The study collected demographic data, medical histories, comprehensive ophthalmologic examination results, and HbA1c levels from routine laboratory records for all participants.

## Peripheral venous blood collection

After a clinical evaluation by a specialist, participants proceeded to the blood collection unit for serum ELA level assessment. Venous blood was drawn from the antecubital vein using yellow-capped tubes following a fasting period of 8–12 h. The collected samples were allowed to clot for 20 min, then centrifuged at 4000 RPM for 10 min using an Nf 1200R centrifuge (Nüve, Türkiye). The separated sera were divided into aliquots using eppendorf tubes and preserved at −80 °C until further analysis.

## Measurement of elabela using ELISA

After reaching the target sample size, serum samples stored at −80 °C were thawed at room temperature. The measurement of serum ELA levels was performed using a commercial ELISA kit (EH4533; Wuhan Fine Biotech, Wuhan, China) with a detection range of 31.25–2,000 pg/mL and a sensitivity limit of 18.75 pg/mL. This kit utilizes a sandwich ELISA method with a 96-well plate pre-coated with anti-ELA antibodies.

The assay began with the addition of 100 μL of either standards or serum samples to the wells, followed by 90 min of incubation at 37 °C. After washing away unbound materials, 100 μL of biotin-labeled detection antibody was introduced, and the plate underwent another 60-min incubation at the same temperature. Subsequent washing steps were followed by the addition of 100 μL horseradish peroxidase (HRP)-streptavidin, with a further 30-min incubation at 37 °C. Finally, 90 μL of tetramethylbenzidine (TMB) substrate solution was added, and the reaction was allowed to develop for 10–20 min before being halted with 50 μL of stop solution.

The resulting color change was measured at 450 nm using a Bio-Tek 800TS microplate reader (Bio-Tek Instruments, Winooski, VT, USA). ELA concentrations were quantified by referencing a standard curve constructed from known concentrations.

### Sample size determination

The sample size was calculated using G*Power version 3.1.9.7 software based on the methodology described by *Zhang et al. (2018)*. The parameters were set as follows: a type I error ($\alpha$) rate of 5%, an effect size of 0.550, a critical F value of 2.758, and a non-centrality parameter ($\lambda$) of 19.38. To achieve a statistical power of 95%, a total of 64 participants (16 per group) was required.

### Statistical analysis

Data analyses were performed using SPSS version 22.0 statistical software (IBM SPSS; Armonk, NY, USA). The Shapiro-Wilk test was employed to assess the normality of the data. Continuous variables were reported as mean and standard deviation (SD), while categorical variables (such as sex) were presented as counts. Continuous data were analyzed using the Kruskal-Wallis test and one-way ANOVA, with *post hoc* multiple comparisons performed using the Dunn-Bonferroni test for Kruskal-Wallis. *P*-values were adjusted for multiple comparisons. Categorical data were evaluated with the chi-square test. Spearman correlation coefficients were calculated. A significance level of $P \leq 0.05$ was used.

## RESULTS

A total of 60 DM patients, 20 with non-DR, 20 with NPDR and 20 with PDR, were included in the study, while twenty patients with without DM were in the control group. Demographic data of the study groups are presented in Table 1. In our study, the age, sex, and body mass index of the between groups were similar ($p = 0.905$, $0.985$ and $0.241$, respectively). DM durations were found to be significantly higher in the PDR group than in the non-DR group between DM subgroups ($p = 0.002$). There was a no statistically significant difference in the HbA1c levels between DM subgroups ($p = 0.199$). Serum ELA levels were 217.19 ± 97.54 pg/mL in the non-DR group, 221.76 ± 93.12 pg/mL in the NPDR group, 302.35 ± 146.17 pg/mL in the PDR group and 216.49 ± 58.85 pg/mL in the control group. Serum ELA levels were no statistically significant difference in the between the DM subgroups and the control group and the DM patients and the control group (Fig. 1 and Table 1).

In the correlation analysis, there wasn't a correlation between the serum ELA levels and HbA1c levels, DM durations, age and body mass index (BMI). The results of the correlation analysis of the serum ELA levels with the HbA1c levels, DM durations, age and BMI values are presented in Table 2 and Fig. 2. Additionally, the within-group correlation between ELA levels and other parameters was analyzed in the subgroups. In the non-DR and NPDR groups, no correlation was found between ELA levels and HbA1c levels, DM duration, age, or BMI parameters. However, in the control and PDR groups, a weak to

**Table 1 The results of the blood samples and the demographic data of the study.**

| Parameters | Control | non-DR | NPDR | PDR | P |
|---|---|---|---|---|---|
| Age (years)** | 65.65 ± 8.01 | 65.50 ± 7.74 | 66.90 ± 5.99 | 65.15 ± 9.42 | 0.905[a] |
| Sex (Male/Female)** | 8/12 | 8/12 | 8/12 | 9/11 | 0.985 |
| BMI (kg/cm$^2$)** | 28.71 ± 4.78 | 29.49 ± 3.68 | 30.45 ± 4.08 | 27.87 ± 4.78 | 0.241[a] |
| Elabela (pg/mL)** | 216.49 ± 58.85 | 217.19 ± 97.54 | 221.76 ± 93.12 | 302.35 ± 146.17 | 0.123[b] |
| 95% Cl for Elabela mean | [192.45–242.82] | [182.22–263.33] | [184.76–261.27] | [240.46–363.69] | |
| 95% Cl for Elabela standard deviation | [36.43–76.45] | [42.51–144.36] | [46.83–126.19] | [93.87–180.04] | |
| HbA1c (%)** | | 8.19 ± 2.30 | 9.30 ± 1.96 | 9.15 ± 1.90 | 0.199[a] |
| DM durations (years)** | | 7.12 ± 7.49 | 8.91 ± 5.92 | 14.53 ± 5.46 | 0.001[*b] |

Notes:
[*] Significance level: $p < 0.05$.
[**] Values were expressed as means ± standard deviation.
[a] One-way ANOVA test was applied.
[b] Kruskal-Wallis test was applied.
BMI, Body Mass Index; Cl, confidence interval; DM, Diabetes Mellitus; non-DR, without diabetic retinopathy; NPDR, Non-proliferative diabetic retinopathy; PDR, proliferative diabetic retinopathy.

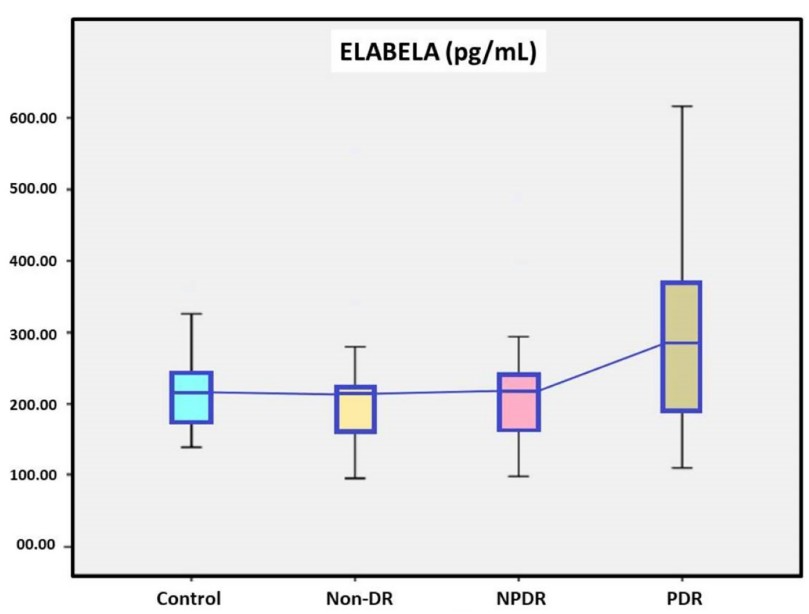

**Figure 1 The serum elabela levels of study groups.**

moderate correlation was observed only between ELA levels and age ($p = 0.015$, r = −0.536, and $p = 0.032$, r = 0.481).

## DISCUSSION

This study explored the potential role of serum ELA levels in DR. Although serum ELA levels were higher in diabetic patients than in controls, the differences did not reach statistical significance. Subgroup analysis indicated that ELA levels were elevated in the PDR group (302.35 ± 146.17 pg/mL) compared to the non-DR group (217.19 ± 97.54 pg/mL) and the NPDR group (221.76 ± 93.12 pg/mL), but these differences were also not statistically
**Table 2 The results of the correlation analysis of the serum elabela levels with the HbA1c levels, DM durations, age and BMI values.**

|  | Statistics | HbA1c | DM durations | Age | BMI |
|---|---|---|---|---|---|
| **Serum elabela levels** | Correlation coefficient | 0.009 | 0.219 | 0.083 | −0.130 |
|  | P value | 0.945 | 0.095 | 0.464 | 0.250 |
|  | n | 60 | 60 | 80 | 80 |

Note:
BMI, Body Mass Index; DM, Diabetes Mellitus.

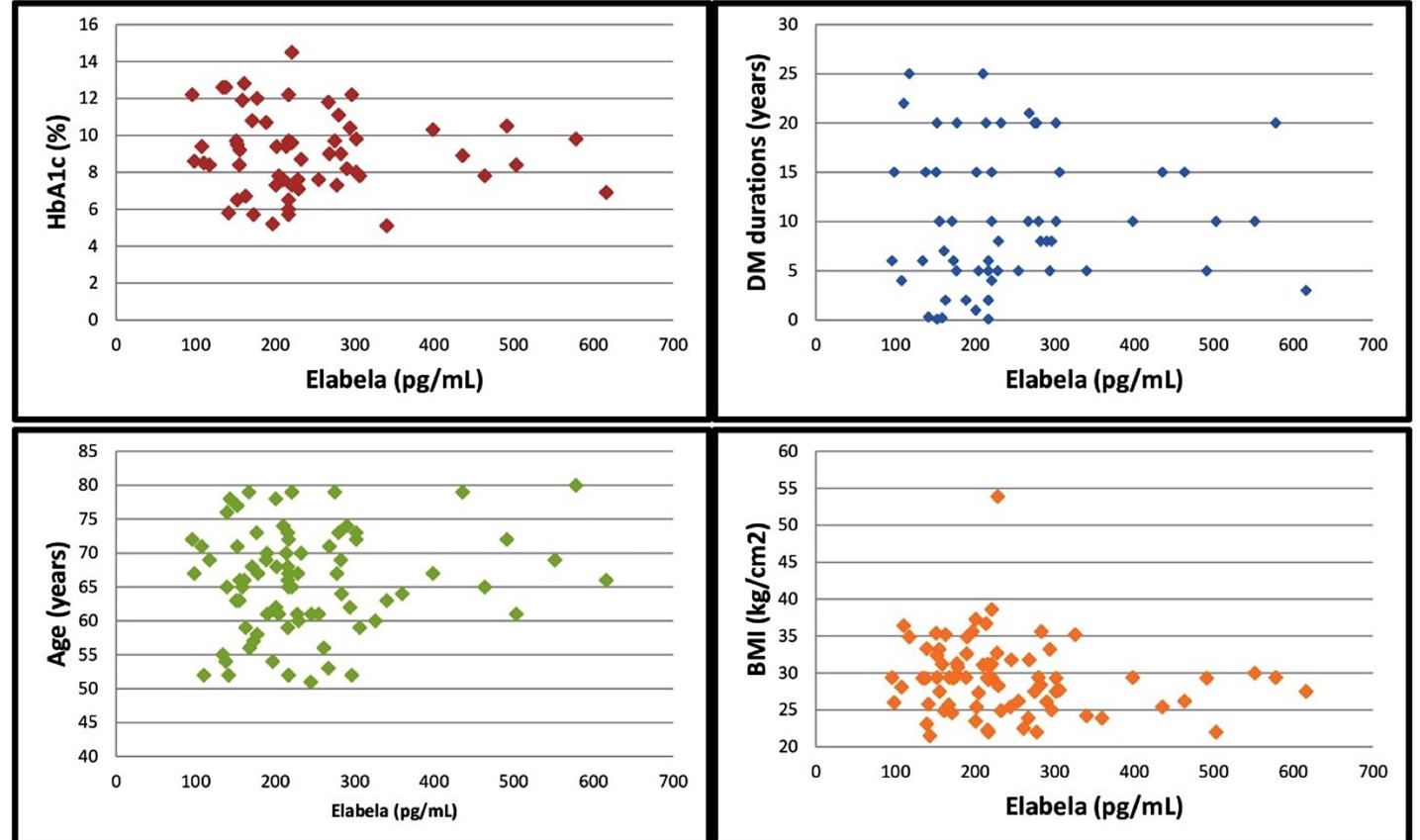

**Figure 2 Correlation of serum elabela levels with HbA1c levels, DM duration, age and BMI values.**

significant. Additionally, DM durations were found to be significantly higher in the PDR group than in the non-DR group between DM subgroups (Table 1).

The pathophysiology of diabetic retinopathy is complex and involves several key disorders, primarily related to disruptions in glucose homeostasis, insulin secretion, and action (*Wan et al., 2015*). Elevated blood glucose levels contribute to retinal ischemia, hypoxia, and the growth of abnormal blood vessels, which ultimately lead to retinopathy (*van der Giet et al., 2015*). Recent research suggests that DR is not only a microvascular complication of diabetes but also a neurodegenerative condition. DR results from dysfunction in the retinal neurovascular unit, which involves cell death, inflammatory

responses, neovascular growth, and disruption of the blood-retina barrier (*Wang & Lo, 2018*; *Öhman et al., 2018*). As a result, the ADA now classifies DR as a specific neurovascular complication of both type 1 and type 2 diabetes (*Flaxel et al., 2020*). Observational studies have identified dyslipidemia, mitochondrial apoptosis, and oxidative stress as major pathological changes in DR (*Miller, Cascio & Rosca, 2020*; *Fort et al., 2021*; *Lin, Qin & Guo, 2022*; *Rao et al., 2021*). The retina, a complex tissue with at least 10 distinct layers (*Miller, Cascio & Rosca, 2020*), consists of photoreceptor cells, retinal ganglion cells, bipolar cells, amacrine cells, horizontal cells, glial cells, endothelial cells (EC), pericytes, and retinal pigment epithelial (RPE) cells. Diabetes disrupts the normal interactions between these cells, leading to significant damage and the loss of nearly all retinal cell types (*Szabó et al., 2017*). Pericytes, which are the first vascular cells affected by diabetes, are notably lost in early NPDR, impairing the retinal vascular microcirculation (*Arboleda-Velasquez et al., 2015*). Despite considerable progress, the exact mechanisms behind DR remain unclear, and ongoing studies continue to explore the various factors and pathways involved.

Research into circulating biomarkers has the potential to enhance our understanding of DR and pave the way for innovative clinical strategies. With this in mind, our study sought to explore the relationship between ELA levels and DR. Our study provides valuable insights into the potential role of ELA levels in DR. Although the differences in ELA levels were not statistically significant, they are noteworthy. The lack of statistical significance in the observed increase in ELA levels, particularly in the PDR group, may be attributed to the small sample sizes within the subgroups and the division into a large number of subgroups. Future studies could use the ELA levels reported in our study as a reference in statistical analyses to achieve meaningful results in larger patient populations. Additionally, our study confirmed a positive correlation between the duration of diabetes and the increasing incidence of DR (*Sasso et al., 2022*). A review of the literature revealed only one study investigating ELA levels in DR. This study, presented as a poster at the 82nd Scientific Sessions of the American Diabetes Association, divided type 2 DM patients into three groups: non-DR, NPDR, and PDR. It was reported that ELA levels decreased gradually from the non-DR group to the PDR group, with statistically significant differences among the three groups. Additionally, the study found that the duration of diabetes in the non-DR group was shorter than in the NPDR and PDR groups (*Gu et al., 2022*).

Although there is only one similar study on ELA in DR, several studies have explored its role in other conditions. *Chen et al. (2017)* reported a significant improvement in acute kidney injury with elabela therapy. *Onalan et al. (2020)* observed a marked decrease in serum ELA levels in DKD patients as the severity of the disease increased. In another study, the authors noted that ELA levels were significantly correlated with albuminuria and served as a clinically relevant predictor for DKD patients (*Zhang et al., 2018*). *Guo et al. (2020)* observed a decrease in ELA levels during the second trimester in patients with gestational diabetes mellitus (GDM) and suggested a possible link between impaired ELA secretion and GDM pathogenesis, highlighting the need for further research to clarify this relationship. Similarly, *Karagoz et al. (2022)* found lower ELA levels in GDM patients

compared to controls. Collectively, these studies suggest that elabela may have a protective role in DKD and GDM.

Before ELA was discovered, it was widely believed that APLN was the sole ligand for the G protein-dependent APJ receptor (*Zhang et al., 2018*). However, recent studies have shown that ELA and APLN have similar functions and both interact with the APJ receptor, forming a crucial signaling pathway in various physiological processes (*Kuba et al., 2019*). The APLN/APJ system is involved in regulating diseases of organs like the lungs, kidneys, liver, and muscles (*Yan et al., 2020*; *Huang, Wu & Chen, 2018*; *Lv et al., 2017*; *Luo et al., 2021*). It is also strongly linked to the pathophysiological processes of various cell types, including endothelial cells (ECs), vascular smooth muscle cells, and cardiomyocytes (*Cheng et al., 2019*; *Luo et al., 2018*; *Lu et al., 2017*). Specifically, the APLN/APJ system plays a key role in the development of eye diseases. Elevated serum APLN levels are found in patients with age-related macular degeneration (*Vural et al., 2021*), while low levels are observed in those with exfoliative glaucoma (*Bezci Aygun et al., 2020*). More crucially, the APLN/APJ system influences eye health by impacting the retina (*Baden, Euler & Berens, 2020*). It is highly expressed in vascular ECs, where it regulates vital processes such as EC proliferation, migration, and angiogenesis (*Cheng et al., 2019*). Retinal EC proliferation is vital for healthy eye development (*Aguilar et al., 2008*), while abnormal EC proliferation leads to pathological angiogenesis and retinopathy (*Li, Wang & Zhang, 2015*; *Carmeliet & Tessier-Lavigne, 2005*). Pathological angiogenesis contributes to several retinal diseases, including retinopathy of prematurity, DR, central retinal vein occlusion, and age-related macular degeneration (*Li, Wang & Zhang, 2015*). Numerous studies have investigated the role of apelin, one of the APJ ligands, in the development of diabetic retinopathy. APLN has been reported to contribute to the proliferation of retinal endothelial cells and the angiogenesis process (*Soualmia et al., 2017*). *Dawood et al. (2016)* found that APLN levels were significantly higher in patients with PDR compared to those with NPDR and non-DR and suggested that APLN plays a role in the progression of PDR. *Yasir et al. (2022)* suggested that APLN could independently assess the severity of DR and serve as a distinguishing marker for retinopathy. A meta-analysis examining the link between DR and various adipocytokines reported that APLN levels were higher in DR patients compared to controls, though the difference was not statistically significant. However, the studies reviewed in this meta-analysis article show varying results regarding the relationship between APLN and DR (*Jiang et al., 2023*). *Feng et al. (2022)* suggested that apelin overexpression in the early stages of DR has a protective effect on pericytes, preventing the breakdown of the blood-retinal barrier, and proposed APLN as a potential treatment for early-stage DR. Despite the significant focus on APLN's role in DR pathogenesis, we believe further research is needed to explore the complexities of APLN's role in DR, given the heterogeneous findings. Similarly to the heterogeneity of findings in APLN studies in DR, our study shows heterogeneous results compared to the only existing study in the literature examining ELA levels in DR. This highlights the need for further research to better understand the relationship between ELA and DR. Additionally, while the elevation of ELA levels in the PDR group did not reach statistical significance in our study, we believe that, when considered in conjunction with existing literature on APLN—

another ligand of the APJ receptor—ELA may play a potential role in endothelial cell proliferation and angiogenesis in the pathogenesis of DR.

A cross-sectional design was chosen in this study due to its practicality and feasibility in capturing baseline data, as well as its ability to simultaneously evaluate serum ELA levels at different stages of DR progression, allowing for the immediate understanding of potential correlations between disease severity and biomarker expression. While cross-sectional designs inherently lack the ability to establish causality or track temporal changes, they serve as a robust starting point for exploring novel biomarkers like ELA. Given the limited prior research on ELA in the context of DR, this design was deemed most suitable for hypothesis generation and initial validation. We recognize that a longitudinal study would provide further insights into the dynamic changes of ELA levels over time and their potential role in predicting DR progression. However, the cross-sectional approach adopted here establishes the foundational relationship between ELA levels and DR severity, paving the way for future longitudinal or interventional research. Importantly, the lack of prior data on ELA in this field underscores the value of a cross-sectional design to guide subsequent, more resource-intensive studies by identifying key variables and relationships warranting deeper exploration. Our study is a preliminary investigation into the role of ELA in DR and while statistical significance was not achieved, the observed trends, especially in the PDR group, point to possible biological relevance. Despite calculating the sample size using appropriate statistical methods, the sample size may have been insufficient for this particular study. A larger patient group, might have shown significant results in relation to ELA levels, especially within the PDR subgroup, and we acknowledge this as a limitation. Small sample sizes can reduce statistical power, making it more challenging to detect subtle but biologically relevant differences. Additionally, variability in biomarker levels among individuals can obscure group-level differences. Future research should consider these challenges by including larger sample sizes to improve consistency and reliability. Additionally, our study was single-center. Furthermore, data on potential confounding factors such as hypertension, smoking, medication use, and other relevant factors were not included in the study. This represents a potential limitations of the study. Stratification by additional variables, such as hypertension, medication use, lifestyle factors, and presence of comorbid conditions, could further clarify the relationship between ELA and DR progression.

## CONCLUSIONS

Although the elevated ELA levels in the PDR group did not reach statistical significance, they offer valuable insights for future research on the relationship between DR and ELA. It is also considered important to explore the connection between retinal pathological angiogenesis and ELA, particularly given the observed increase in the PDR patient group. Larger-scale studies with more participants are anticipated to significantly advance our understanding of the pathogenesis of DR.

### Funding
The authors received no funding for this work.

### Competing Interests
The authors declare that they have no competing interests.

### Author Contributions
- Muhammed Seyithanoğlu conceived and designed the experiments, performed the experiments, analyzed the data, prepared figures and/or tables, authored or reviewed drafts of the article, and approved the final draft.
- Selma Meşen conceived and designed the experiments, performed the experiments, analyzed the data, authored or reviewed drafts of the article, and approved the final draft.
- Aysegul Comez conceived and designed the experiments, performed the experiments, authored or reviewed drafts of the article, and approved the final draft.
- Ali Meşen conceived and designed the experiments, performed the experiments, authored or reviewed drafts of the article, and approved the final draft.
- Abdullah Beyoğlu conceived and designed the experiments, performed the experiments, authored or reviewed drafts of the article, and approved the final draft.
- Yaşarcan Baykişi performed the experiments, prepared figures and/or tables, authored or reviewed drafts of the article, and approved the final draft.
- Filiz Alkan Baylan performed the experiments, authored or reviewed drafts of the article, and approved the final draft.

### Human Ethics
The following information was supplied relating to ethical approvals (*i.e.*, approving body and any reference numbers):

Received approval from the Clinical Research Ethics Committee of Kahramanmaraş Sütçü Imam University Faculty of Medicine (2023/17).

### Data Availability
The raw measurements are available in the Supplemental File.

### Supplemental Information
Supplemental information for this article can be found online at http://dx.doi.org/10.7717/peerj.18841#supplemental-information.

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
