# Peer review of "The potential of serum elabela levels as a marker of diabetic retinopathy: results from a pilot cross-sectional study"

_PeerJ, doi:10.7717/peerj.18841_

## Round 0.1 · original submission · Major Revisions

We are pleased to let you know that your manuscript has now passed through the review stage and is ready for revision. I am sorry that it has taken longer than normal to receive review for your manuscript. The reviewers provided detailed comments, and I ask that you consider these carefully and correct the errors when revising the manuscript as well as respond to their suggestions in the cover letter when you re-submit. This will help avoid further rounds of explanations and revisions, and allow quickly move to the decision.

You may find the following literature useful to expand the discussion of the results obtained:

Wensha Gu, Min Shi, Yanna Chen, Yuting Liu, Jing Song, Hong Zhang; 473-P: Correlation between Serum Elabela and Type 2 Diabetic Retinopathy. Diabetes 1 June 2022; 71 (Supplement_1): 473–P. https://doi.org/10.2337/db22-473-P

Dawood, Alaa Eldin Abdl El-salam; Serag, Amany Anwar Mohamed; Ellakwa, Amin Faisal; Elela, Dalia Hosny Abou; and Gazareen, Sanaa Sayed (2016) "The role of serum apelin in diabetic patients with retinopathy," Menoufia Medical Journal: Vol. 29: Iss. 1, Article 5.
DOI: https://doi.org/10.4103/1110-2098.178940

Luo J, Zhao Q, Li Z, Chen L. Multiple roles of apelin/APJ system in eye diseases. Peptides. 2022 Jun;152:170767. doi: 10.1016/j.peptides.2022.170767.

Reviewer 1 ·

Basic reporting

This study analyzed the differences of Serum elabela levels in no DM, no DR, NPDR and PDR groups, attempting to discuss the potential association between serum creatinine and DR progression. However, the p value of this study showed no significant difference in serum creatinine levels among different groups. The authors explain and analyze this in the discussion section, suggesting that Serum elabela levels may be a potential marker of DR.

Experimental design

1. For the classification of DR (line72-75), please give a specific description: what kind of examinations method were used, fundus photography? OCT? Indirect fundoscopy? If using manual examination, is it checked by two or more ophthalmologists? Please give details in the text.
2. Was there an ethical code for this study? If so, please add it to the paper (line63-66).

Validity of the findings

The results of this study found that Serum elabela levels were not statistically significant among the groups, as well as the associations between Serum elabela levels and the indicators were also not statistically significant, but the authors insisted in the conclusions that the potential role of Serum elabela in the progression of DR. Please provide a detailed analysis of these results in the discussion section and explain the possible reasons for the results.

Additional comments

Abstract
1. Paragraph formatting is recommended to distribute the text evenly between the margins.
2. Please check the text for full and short forms of proper nouns. Please use abbreviations uniformly except for the first time.

Result
1. Were the p values in Table 1 subject to multiple corrections? Were other confounding factors, such as hypertension, smoking, education level, etc. taken into account between the groups?
2.Table 1 misspelling (line279) with “non-Dr”. Please check the text for other grammatical or spelling errors.
3. Was the correlation analysis in Table 2 considered in different subgroups?

Discussion
1. Please go deeper into the discussion of the mechanism section and add the literature:
1) Ren J, Zhang S, Pan Y, et al. Diabetic retinopathy: Involved cells, biomarkers, and treatments. Front Pharmacol. 2022;13:953691. 
2) Wu J, Kim YJ, Dacey DM, Troy JB, Smith RG. Two mechanisms for direction selectivity in a model of the primate starburst amacrine cell. Visual Neuroscience. 2023;40:E003.

2. Please add an explanation of the limitations of this study at the end of the discussion.

·

Basic reporting

no comment

Experimental design

no comment

Validity of the findings

no comment

Additional comments

I congratulate the authors for this interesting paper which gives a valuable addition to the advancement in the field of diabetic retinopathy research.
The research is novel which needs to be validated by a larger study to confirm the results of the study.

Reviewer 3 ·

Basic reporting

Title & Abstract
Title
The title clearly identifies the condition under study and the overall aim of the research. The authors might consider adding the study design, for example: “The potential of serum elabela levels as a marker of diabetic retinopathy: results from a pilot cross-sectional study”.
Identifying the study as a pilot is important to give the right impression about the level of conclusions that can be reached. This will better predispose the readers of the article to the contents.
Abstract
The abstract is structured adequately. Some suggestions for improvements:
• Page 5 line 18: please use the abbreviation for elabela (i.e., ELA) consistently throughout the text.
• Page 5 line 29: please state that there were no differences in age, gender, and HB1Ac levels between controls and DM patients.
• Page 5 line 30: please provide the figures for ELA levels for the groups under analysis (i.e., controls, non-DR, NPDR, and PDR).

Introduction
Diabetic retinopathy (DR) is a major complication of diabetes, becoming more likely as the disease duration increases. DR has two stages: non-proliferative (NPDR) and proliferative (PDR). Elabela (ELA) is a peptide active in early development and circulates in adults as a hormonal peptide. It binds to the apelin receptor (APJ). The apelinergic pathway, involved in cardiovascular and metabolic diseases, has gained attention for its therapeutic potential. ELA affects blood vessel relaxation and blood pressure, and its role in metabolism is under investigation.
There are no reports about the relationship between ELA and DR, and thus this study is valuable, as it is the first of its kind. The objective is clearly defined.
We have some suggestions for improvements:
• Page 6 Line 57: the authors might also want to mention that the apelin/APJ system is involved in many eye diseases (https://doi.org/10.1016/j.peptides.2022.170767).
• Page 6 Line 58: The authors might want to consider briefly reviewing the evidence of the involvement of ELA in renal complications of diabetes (for example: https://doi.org/10.1159/000492093 and https://doi.org/10.4314/ahs.v20i2.37). This will strengthen the support for the hypothesis.

Overall, the language of the manuscript is clear and unambiguous. The authors used professional English throughout the text.

Figures & Tables
Tables are adequate and they summarise the content effectively.

Experimental design

The methodology for collecting blood samples and determining ELA blood levels are adequate. Conversely, the description of the sample could be improved.
Suggestions for improvements:
• Page 9 Line 69: please add information about how patients were diagnosed with DM.
This study cannot conclude on the temporal relationship between DR and ELA, due is cross-sectional design. However, the authors are not making any causality claim, and this is a minor detail.
The authors performed a sample size analysis before conducting the study. The target effect size was 0.855, which is considered as large. In the absence of previous evidence, it is not clear how this effect size was chosen.
Suggestions for improvements:
• Page 7 Line 113: please provide more data to justify the effect size chosen. Why was data from Yener et al chosen? Perhaps data coming from studies about the relationship between ELA and diabetic nephropathy would have been more accurate. Please justify your choice.
This is a well-designed cross-sectional study. Regrettably, the study was probably underpowered to fulfill its objective. However, results are still encouraging and relevant to the field.

Validity of the findings

Results
There were no major differences in the demographic and disease characteristics of the groups under study. The differences in ELA levels between the PDR and the rest of the groups is noteworthy, even if it did not reach statistical significance.
In the absence of any previous evidence, the results are encouraging and relevant to the field, as mentioned earlier.
Suggestions for improvement:
• Page 8 Line 126: do you have data about the participants’ medications? It would be important to mention them here.

Discussion
The discussion is concise and adequate for the results of the study.
Suggestions for improvement:
• Page 8 Line 140: please add a starting paragraph summarizing the main results of the study, which are mentioned in lines 150-154.
• Page 9 Line 186: please add a paragraph mentioning study limitations. It is important to discuss the issues with statistical power.

Conclusion
The conclusion does not overstate the importance of findings, which is essential for the credibility of the study. Possible future investigations are discussed.

Additional comments

no comment

---

## Round 0.2 · Minor Revisions

Your manuscript has now been evaluated. There is still a request for minor changes, before I can accept the manuscript. Please consider the comments carefully and submit the final version as well as respond to the suggestions in the cover letter when you re-submit.

Reviewer 3 ·

Basic reporting

Title & Abstract
Title
The revision is adequate.
Abstract
The abstract is well written. All the mentioned suggestions are adequately addressed.

Introduction
The section is well written and the mentioned suggestions are addressed. Please see the below suggestions as well and make the required revisions:
• The introduction briefly mentions the stages of diabetic retinopathy (non-proliferative and proliferative) but does not delve into the underlying mechanisms, such as vascular dysfunction, inflammation, or oxidative stress, which are relevant for biomarker studies.
For example, “The interplay between vascular and neuronal damage in DR. How hypoxia drives neovascularization in PDR”. Connect these mechanisms to the potential role of biomarkers like ELA in detecting or monitoring DR progression.
• The introduction briefly cites one prior study on ELA and DR but could expand on the findings or gaps in other studies to provide a stronger justification for the current research. The research question or hypothesis is not clearly stated, which could help readers immediately understand the study’s purpose.
For example, "This study aims to measure serum ELA levels in DR patients and may serve as a potential biomarker for DR progression."

Figures & Tables
This is adequate.

Experimental design

Material and Methods
The authors have addressed the mentioned suggestions. Please see some additional suggestions as below:
• The cross-sectional design provides a snapshot of ELA levels but does not capture temporal changes or causality and the study does not justify why this approach was chosen over others (e.g., case-control or longitudinal designs). We suggest that a longitudinal approach could provide more meaningful insights into the relationship between ELA levels and DR progression.
Suggestion: Include a rationale for selecting a cross-sectional design and discuss its limitations in observing causality or temporal changes. A longitudinal approach could provide more meaningful insights into the relationship between ELA levels and DR progression.
• The recruitment process (e.g., inclusion/exclusion based on medical records or self-report) is not clearly detailed. Some exclusions (e.g., recent medication use or other systemic diseases) lack granularity regarding how these were assessed (self-report, medical records, etc.).
Suggestion: Specify methods for verifying participants’ medical history and the recruitment setting (e.g., random sampling or broad inclusion criteria, outpatient clinics) to reduce potential bias.
• Study period is not mentioned in the method section. Please mention it.

Validity of the findings

Results
• As mentioned previously, the results only report means and standard deviations for ELA levels and it did not reach statistical significance. Please include confidence intervals (CIs), which would provide a better sense of variability and precision.
• Also, the authors can consider adding visual aids such as: Box plots to compare ELA levels across subgroups. Correlation plots for ELA levels vs. HbA1c, diabetes duration, or other parameters. A summary figure showing trends in ELA levels across DR stages.

Discussion
The revisions are adequate. In addition, the authors can also consider addressing the implications of the lack of statistical significance in the results by explain how the small sample size, variability in ELA levels, or methodological constraints might have influenced the results. You can also add adjustments for future studies, such as larger sample sizes, stratification by additional variables, or refined biomarker measurement techniques.
The section is overall well-written.

Conclusion
The conclusion is adequate and does not overstate the importance of findings.

Additional comments

None

---

## Round 0.3 · accepted · Accept

In the revised version the authors took into consideration all comments and remarks. I recommend to accept the manuscript for publication in PeerJ.